# Copulas as High-Dimensional Generative Models: Vine Copula Autoencoders

**Natasa Tagasovska**
Department of Information Systems
HEC Lausanne, Switzerland
natasa.tagasovska@unil.ch

**Damien Ackerer**
Swissquote Bank
Gland, Switzerland
damien.ackerer@swissquote.ch

**Thibault Vatter**
Department of Statistics
Columbia University, New York, USA
thibault.vatter@columbia.edu

## Abstract

We introduce the vine copula autoencoder (VCAE), a flexible generative model for high-dimensional distributions built in a straightforward three-step procedure. First, an autoencoder (AE) compresses the data into a lower dimensional representation. Second, the multivariate distribution of the encoded data is estimated with vine copulas. Third, a generative model is obtained by combining the estimated distribution with the decoder part of the AE. As such, the proposed approach can transform any already trained AE into a flexible generative model at a low computational cost. This is an advantage over existing generative models such as adversarial networks and variational AEs which can be difficult to train and can impose strong assumptions on the latent space. Experiments on MNIST, Street View House Numbers and Large-Scale CelebFaces Attributes datasets show that VCAEs can achieve competitive results to standard baselines.

## 1   Introduction

Exploiting the statistical structure of high-dimensional distributions behind audio, images, or video data is at the core of machine learning. Generative models aim not only at creating feature representations, but also at providing means of sampling new realistic data points. Two classes are typically distinguished: *explicit* and *implicit* generative models. Explicit generative models make distributional assumptions on the data generative process. For example, *variational autoencoders* (VAEs) assume that the latent features are independent and normally distributed [37]. Implicit generative models make no statistical assumption but leverage another mechanism to transform noise into realistic data. For example, *generative adversarial networks* (GANs) use a discriminant model penalizing the loss function of a generative model producing unrealistic data [22]. Interestingly, *adversarial autoencoders* (AAEs) combined both features as they use a discriminant model penalizing the loss function of an encoder when the encoded data distribution differs from the prior (Gaussian) distribution [48]. All of these new types of generative models have achieved unprecedent results and also proved to be computationally more efficient than the first generation of deep generative models which require Markov chain Monte Carlo methods [32, 30]. However, adversarial approaches require multiple models to be trained, leading to difficulties and computational burden [62, 26, 24], and variational approaches make (strong) distributional assumptions, potentially detrimental to the generative model performance [64].

We present a novel approach to construct a generative model which is simple, makes no prior distributional assumption (over the input or latent space), and is computationally efficient: the *vine copula autoencoders* (VCAEs). Our approach, schematized in Figure 1 combines three tasks. First, an

autoencoder (AE) is trained to provide high-quality embeddings of the data. Second, the multivariate distribution of the encoded train data is estimated with vine copulas, namely, a flexible tool to construct high-dimensional multivariate distributions [3, 4, 1]. Third, a generative model is obtained by combining the estimated vine copula distribution with the decoder part of the AE.

In other words, new data is produced by decoding random samples generated from the vine copula. An already trained AE can thus be transformed into a generative model, where the only additional cost would be the estimation of the vine copula. We show in multiple experiments that this approach performs well in building generative models for the MNIST, Large-Scale CelebFaces Attributes, and Street View House Numbers datasets. To the best of our knowledge, this is the first time that vine copulas are used to construct generative models for very high dimensional data (such as images).

Next, we review the related work most relevant to our setting. The most widespread generative models nowadays focus on synthetic image generation, and mainly fall into the GAN or VAE categories, some interesting recent developments include [49, 15, 26, 76, 29, 14, 6]. These modern approaches have been largely inspired by previous generative models such as belief networks [32], independent component

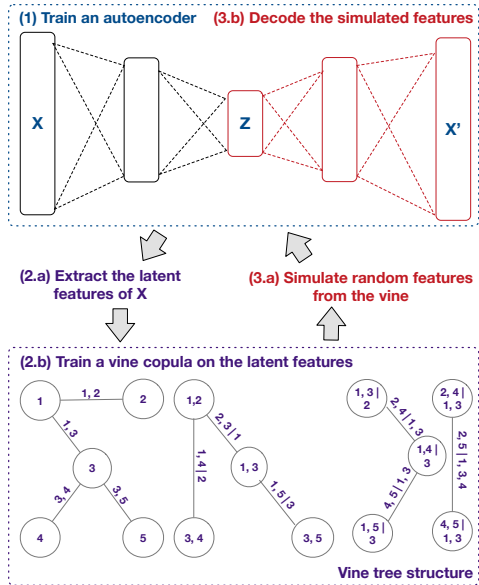

Figure 1: Conceptual illustration of a VCAE.

analysis [33] or denoising AEs [79]. Part of their success can be attributed to the powerful neural network architectures which provide high quality feature representations, often using Convolutional architectures [41]. A completely different framework to model multivariate distributions has been developed in the statistical literature: the so-called *copulas*. Thanks to their ability to capture complex dependence structures, copulas have been applied to a wide range of scientific problems, and their successes have led to continual advances in both theory and open-source software availability. We refer to [56, 35] for textbook introductions. More recently, copulas also made their way into machine learning research [43, 20, 47, 78, 45, 13, 74, 38]. However, copulas have not yet been employed in constructing high dimensional generative models. While [42, 59] use copulas for synthetic data generation, they rely on strong parametric assumptions. In this work, we illustrate how *nonparametric vine copulas* allow for arbitrary density estimation [50], which in turn can be used to sample realistic synthetic datasets.

Because their training is relatively straightforward, VCAEs have some advantages over GANs. For instance, GANs require some complex modifications of the baseline algorithm in order to avoid mode collapse, whereas vines naturally fit multimodal data. Additionally, while GANs suffer from the "exploding gradients" phenomenon (e.g., see [24]) and require careful monitoring of the training and early stopping, this is not an issue with VCAEs as they are built upon standard AEs.

To summarize, the contribution of this work is introducing a novel, competitive generative model based on copulas and AEs. There are three main advantages of the proposed approach. First, it offers modeling flexibility by avoiding most distributional assumptions. Second, training and sampling procedures for high-dimensional data are straightforward. Third, it can be used as a plug-in allowing to turn any AE into generative model, simultaneously allowing it to serve other purposes (e.g., denoising, clustering).

The remainder of the paper is as follows. Section 2 reviews vine copulas as well as their estimation and simulation algorithms. Section 3 discusses the VCAE approach. Section 4 presents the results of our experiments. Section 5 concludes and discusses future research. The supplementary material contains further information on algorithm and experiments, as well as additional experiments.

## 2 Vine copulas

### 2.1 Preliminaries and motivation

A *copula*, from the latin word *link*, flexibly "couples" marginal distributions into a joint distribution. As such, copulas allow to construct joint distributions with the same margins but different dependence structures, or conversely by fixing the dependence structure and changing the individual behaviors. Thanks to this versatility, there has been an exponentially increasing interest in copula-based models over the last two decades. One important reason lies in the following theorem.

**Theorem 1** (Sklar's theorem [71]). *The continuous random vector $\boldsymbol{X} = (X_1, \ldots, X_d)$ has joint distribution $F$ and marginal distributions $F_1, \ldots, F_d$ if and only if there exist a unique copula* [1] *$C$, which is the joint distribution of $\boldsymbol{U} = (U_1, \ldots, U_d) = \big(F_1(X_1), \ldots, F_d(X_d)\big)$.*

Assuming that all densities exist, we can write $f(x_1, \ldots, x_d) = c\{u_1, \ldots, u_d\} \times \prod_{k=1}^{d} f_k(x_k)$, where $u_i = F_i(x_i)$ and $f, c, f_1, \ldots, f_d$ are the densities corresponding to $F, C, F_1, \ldots, F_d$ respectively. As such, copulas allow to decompose a joint density into a product between the marginal densities $f_i$ and the dependence structure represented by the copula density $c$.

This has an important implication for the estimation and sampling of copula-based marginal distributions: algorithms can generally be built into two steps. For instance, estimation is often done by estimating the marginal distributions first, and then using the estimated distributions to construct pseudo-observations via the probability integral transform before estimating the copula density. Similarly, synthetic samples can be obtained by sampling from the copula density first, and then using the inverse probability integral transform to transform the copula sample back to the natural scale of the data. We give a detailed visual example of both the estimation and sampling of (bivariate) copula-based distributions in Figure 2. We also refer to Appendix A.1 or the textbooks [56] and [35] for more detailed introductions on copulas.

The availability of higher-dimensional models is rather limited, yet there exists numerous parametric families in the bivariate case. This has inspired the development of hierarchical models, constructed from cascades of bivariate building blocks: the *pair-copula constructions* (PCCs), also called *vine copulas*. Thanks to its flexibility and computational efficiency, this new class of simple yet versatile models has quickly become a hot-topic of multivariate analysis [2].

### 2.2 Vine copulas construction

Popularized in [3, 4, 1], PCCs model the joint distribution of a random vector by decomposing the problem into modeling pairs of conditional random variables, making the construction of complex dependencies both flexible and yet tractable. Let us exemplify such constructions using a three dimensional vector of continuously distributed random variables $X = (X_1, X_2, X_3)$. The joint density $f$ of $X$ can be decomposed as

$$f = f_1\, f_2\, f_3\, c_{1,2}\, c_{2,3}\, c_{1,3|2}, \tag{1}$$

where we omitted the arguments for the sake of clarity, and $f_1, f_2, f_3$ are the marginal densities of $X_1, X_2, X_3$, $c_{1,2}$ and $c_{2,3}$ are the joint densities of $(F_1(X_1), F_2(X_2))$ and $(F_2(X_2), F_3(X_3))$,

$c_{1,3|2}$ is the joint density of $(F_{1|2}(X_1|X_2), F_{3|2}(X_3|X_2))|X_2$.

The above decomposition can be generalized to an arbitrary dimension $d$ and leads to tractable and flexible probabilistic models [34, 3, 4]. While a decomposition is not unique, it can be organized as a graphical model, a sequence of $d-1$ nested trees, called *regular vine*, *R-vine*, or simply *vine*. Denoting $T_m = (V_m, E_m)$ with $V_m$ and $E_m$ the set of nodes and edges of tree $m$ for $m = 1, \ldots, d-1$, the sequence is a vine if it satisfies a set of conditions guaranteeing that the decomposition leads to a *valid joint density*. The corresponding tree sequence is then called the *structure* of the PCC and has important implications to design efficient algorithms for the estimation and sampling of such models (see Section 2.3 and Section 2.4).

Each edge $e$ is associated to a bivariate copula $c_{j_e,k_e|D_e}$ (a so-called *pair-copula*), with the set $D_e \in \{1, \cdots, d\}$ and the indices $j_e, k_e \in \{1, \cdots, d\}$ forming respectively its *conditioning set* and the *conditioned set*. Finally, the joint copula density can be written as the product of all pair-copula densities $c = \prod_{m=1}^{d-1} \prod_{e \in E_m} c_{j_e,k_e|D_e}$. In the following two sections, we discuss two topics that are

important for the application of vines as generative models: estimation and simulation. For further details, we refer to the numerous books and surveys written about them [16, 39, 72, 18, 2], as well as Appendix A.2.

## 2.3 Sequential estimation

To estimate vine copulas, it is common to follow a sequential approach [1, 27, 50], which we outline below. Assuming that the vine structure is known, the pair-copulas of the first tree, $T_1$, can be directly estimated from the data. But this is not as straightforward for the other trees, since data from the densities $c_{j_e,d_e|D_e}$ are not observed. However, it is possible to sequentially construct "pseudo-observations" using appropriate data transformations, leading to the following estimation procedure, starting with tree $T_1$: for each edge in the tree, estimate all pairs, construct pseudo-observations for the next tree, and iterate. The fact that the tree sequence $T_1, T_2, \ldots, T_{d-1}$ is a regular vine guarantees that at any step in this procedure, all required pseudo-observations are available. Additionally to Appendix A.2.1 and Appendix A.2.2, we further refer to [1, 12, 18, 19, 9, 36] for model selection methods and to [17, 73, 11, 27, 69] for more details on the inference and computational challenges related to PCCs.

Importantly, vines can be truncated after a given number of trees [12, 8, 10] by setting pair-copulas in further trees to independence.

**Complexity**   Because there are $d$ pair-copulas in $T_1$, $d-1$ pair-copulas in $T_2$, ..., and a single pair-copula in $T_{d-1}$, the complexity of this algorithm is $O(f(n) \times d \times \text{truncation level})$, where $f(n)$ is the complexity of estimating a single pair and the truncation level is at most $d-1$. In our implementation, described Section 2.5, $f(n) = O(n)$.

## 2.4 Simulation

Additionally to their flexibility, vines are easy to sample from using inverse transform sampling. Let $C$ be a copula and $U = (U_1, \ldots, U_d)$ is a vector of independent $U(0,1)$ random variables. Then, define $V = (V_1, \ldots, V_d)$ through $V_1 = C^{-1}(U_1)$, $V_2 = C^{-1}(U_2|U_1)$, and so on until $V_d = C^{-1}(U_d|U_1, \ldots, U_{d-1})$, with $C(v_k|v_1, \ldots, v_{k-1})$ is the conditional distribution of $V_k$ given $V_1, \ldots, V_{k-1}$, $k = 2, \ldots, d$. In other words, $V$ is the inverse Rosenblatt transform [65] of $U$. It is then straightforward to notice that $V \sim C$, which can be used to simulate from $C$. As for the sequential estimation procedure, it turns out that

- the fact that the tree sequence $T_1, T_2, \ldots, T_{d-1}$ is a vine guarantees that all the required conditional bivariate copulas are available (see Algorithm 2.2 of [19]),

- the complexity of the algorithm $O(n \times d \times \text{truncation level})$, since $f(n)$ is trivially the complexity required for one inversion multiplied by the number of generated samples.

Furthermore, there exist analytical expressions or good numerical approximations of such inverses for common parametric copula families. We refer to Section 2.5 for a discussion of the inverse computations for nonparametric estimators.

## 2.5 Implementation

To avoid specifying the marginal distributions, we estimate them using a Gaussian kernel with a bandwidth chosen using the direct plug-in methodology of [70]. The observations can then be mapped to the unit square using the probability integral transform (PIT). See steps 1 and 2 of Figure 2 for an example.

Regarding the copula families used as building blocks for the vine, one can contrast parametric and nonparametric approaches. As is common in machine learning and statistics, the default choice is the Gaussian copula. In Section 2.6, we show empirically why this assumption (allowing for dependence between the variables but still in the Gaussian setting) can be too simplistic, resulting in failure to deliver even for three dimensional datasets.

Alternatively, using a nonparametric bivariate copula estimator provides the required flexibility. However, the bivariate Gaussian kernel estimator, targeted at densities of unbounded support, cannot be directly applied to pair-copulas, which are supported in the unit square. To get around this issue, the trick is to transform the data to standard normal margins before using a bivariate Gaussian

kernel. Bivariate copulas are thus estimated nonparametrically using the transformation estimator [67, 47, 50, 21] defined as

$$\widehat{c}(u,v) = \frac{1}{n} \sum_{j=1}^{n} \frac{\mathcal{N}(\Phi^{-1}(u), \Phi^{-1}(v) | \Phi^{-1}(u_j), \Phi^{-1}(v_j), \Sigma)}{\phi\left(\Phi^{-1}(u)\right) \phi\left(\Phi^{-1}(v)\right)}, \qquad (2)$$

where $\mathcal{N}(\cdot, \cdot | v_1, v_2, \Sigma)$ is a two-dimensional Gaussian density with mean $v_1, v_2$, and covariance matrix $\Sigma = n^{-1/3} \operatorname{Cor}(\Phi^{-1}(U), \Phi^{-1}(V))$. For the notation we let $\phi, \Phi$ and $\Phi^{-1}$ to be the standard Gaussian density, distribution and quantile function respectively. See step 3 of Figure 2 for an example.

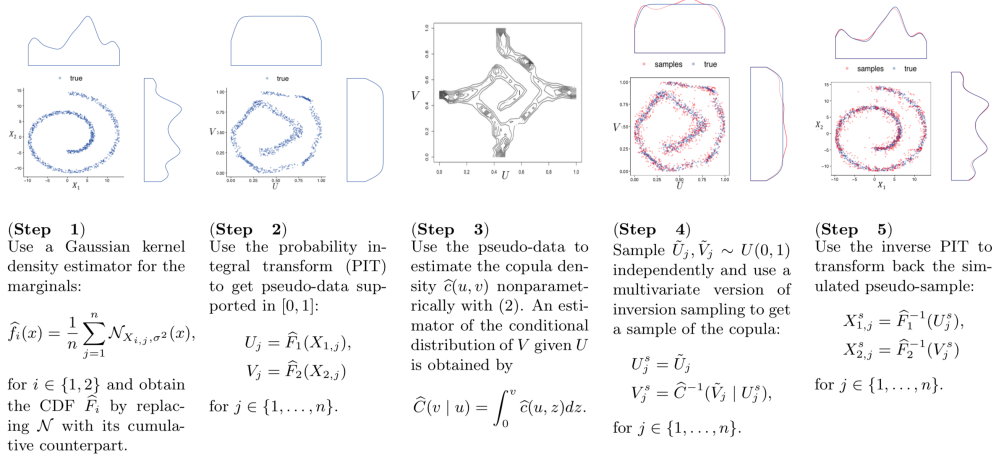

(**Step 1**) Use a Gaussian kernel density estimator for the marginals:

$$\widehat{f}_i(x) = \frac{1}{n} \sum_{j=1}^{n} \mathcal{N}_{X_{i,j}, \sigma^2}(x),$$

for $i \in \{1, 2\}$ and obtain the CDF $\widehat{F}_i$ by replacing $\mathcal{N}$ with its cumulative counterpart.

(**Step 2**) Use the probability integral transform (PIT) to get pseudo-data supported in $[0,1]$:

$$U_j = \widehat{F}_1(X_{1,j}),$$
$$V_j = \widehat{F}_2(X_{2,j})$$

for $j \in \{1, \dots, n\}$.

(**Step 3**) Use the pseudo-data to estimate the copula density $\widehat{c}(u,v)$ nonparametrically with (2). An estimator of the conditional distribution of $V$ given $U$ is obtained by

$$\widehat{C}(v \mid u) = \int_0^v \widehat{c}(u,z)dz.$$

(**Step 4**) Sample $\tilde{U}_j, \tilde{V}_j \sim U(0,1)$ independently and use a multivariate version of inversion sampling to get a sample of the copula:

$$U_j^s = \tilde{U}_j$$
$$V_j^s = \widehat{C}^{-1}(\tilde{V}_j \mid U_j^s),$$

for $j \in \{1, \dots, n\}$.

(**Step 5**) Use the inverse PIT to transform back the simulated pseudo-sample:

$$X_{1,j}^s = \widehat{F}_1^{-1}(U_j^s),$$
$$X_{2,j}^s = \widehat{F}_2^{-1}(V_j^s)$$

for $j \in \{1, \dots, n\}$.

Figure 2: Estimation and sampling algorithm for a pair copula.

Along with vines-related functions (i.e., for sequential estimation and simulation), the Gaussian copula and (2) are implemented in C++ as part of `vinecopulib` [51], a header-only C++ library for copula models based on `Eigen` [25] and `Boost` [68]. In the following experiments, we use the R interface [61] interface to `vinecopulib` called `rvinecopulib` [53], which also include `kde1d` [52] for univariate density estimation.

Note that inverses of partial derivatives of the copula distribution corresponding to (2) are required to sample from a vine, as described in Section 2.4. Internally, `vinecopulib` constructs and stores a grid over $[0,1]^2$ along with the evaluated density at the grid points. Then, bilinear interpolation is used to efficiently compute the copula distribution $\widehat{C}(u,v)$ and its partial derivatives. Finally, `vinecopulib` computes the inverses by numerically inverting the bilinearly interpolated quantities using a vectorized version of the bisection method, and we show a copula sample example as step 4 of Figure 2. The consistency and asymptotic normality of this estimator are derived in [21] under assumptions described in Appendix A.3.

To recover samples on the original scale, the simulated copulas samples, often called pseudo-samples, are then transformed using the inverse PIT, see step 5 of Figure 2. In Appendix C.1, we show that this estimator performs well on two toy bivariate datasets that are typically challenging for GANs: a grid of isotropic Gaussians and the swiss roll.

## 2.6 Vines as generative models

To exemplify the use of vines as generative models, let us consider as a running example a three dimensional dataset $X_1, X_2, X_3$ with $X_1, X_2 \sim U[-5,5]$ and $X_3 = \sqrt{X_1^2 + X_2^2} + U[-0.1, 0.1]$. The joint density can be decomposed as in the right-hand side of (1), and estimated following the procedures described in Section 2.5 and Section 2.3. With the structure and the estimated pair copulas, we can then use vines as generative models.

In Figure 3, we showcase three models. C1 is a nonparametric vine truncated after the first tree. In other words, it sets $c_{2,3|1}$ to independence. C2 is a nonparametric vine with two trees. C3 is a Gaussian vine with two trees. On the left panel, we show their vine structure, namely the trees and the pair copulas. On the right panel, we present synthetic samples from each of the models in blue, with the green data points corresponding to $\sqrt{X_1^2 + X_2^2}$.

Comparing C1 to C2 allows to understand the truncation effect: C2, being more flexible (fitting richer/deeper model), captures better the features of the joint distribution. It can be deduced from the fact that data generated by C2 looks like uniformly spread around the $\sqrt{X_1^2 + X_2^2}$ surface, while data generated by C1 is spread all around. It should be noted that, in both cases, the nonparametric estimator captures the fact that $X_1$ and $X_2$ are independent, as can be seen from the contour densities on the left panel. Regarding C3, it seems clear that Gaussian copulas are not suited to handle this kind of dependencies: for such nonlinearities, the estimated correlations are (close to) zero, as can be seen from the contour densities on the left panel.

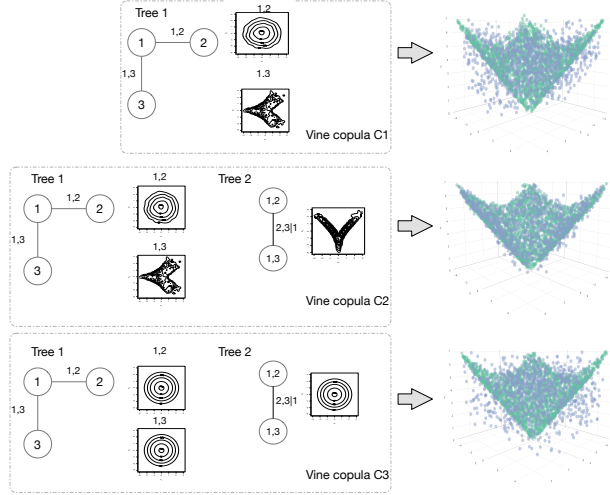

Figure 3: Simulation with different truncation levels, top to bottom - 1 level truncated vine, 2 levels non-parametric vine, 2 levels Gaussian vine.

With this motivation, the next section is dedicated to extending the vine generative approach to high dimensional data. While vines are theoretically suitable for fitting and sampling in high dimensions, they have been only applied to model a few thousands of variables. The reason is mainly that state-of-the-art implementations were geared towards applications such as climate science and financial risk computations. While software such a `vinecopulib` satisfies the requirements of such problems, even low-resolution images (e.g., $64 \times 64 \times 3$) are beyond its current capabilities. To address this challenge, we can rely on the embedded representations provided by neural networks.

## 3  Vine copula autoencoders

The other building block of the VCAE is an *autoencoder* (AE) [7, 31]. These neural network models typically consist of two parts: an *encoder* $f$ mapping a datum $X$ from the original space $\mathcal{X}$ to the latent space $\mathcal{Y}$, and a decoder $g$ mapping a latent code $Y$ from the latent space $\mathcal{Y}$ to the original space $\mathcal{X}$. The AE is trained to reconstruct the original input with minimal reconstruction loss, that is $X' \approx g(f(X))$.

However, AEs simply learn the most informative features to minimize the reconstruction loss, and therefore cannot be considered as generative models. In other words, since they do not learn the distributional properties of the latent features [5], they cannot be used to sample new data points. Because of the latent manifold's complex geometry, attempts using simple distributions (e.g., Gaussian) for the latent space may not provide satisfactory results.

Nonparametric vines naturally fill this gap. After training an AE, we use its encoder component to extract lower dimensional feature representations of the data. Then, we fit a vine without additional restrictions on the latent distribution. With this simple step, we transform AEs into generators, by systematically sampling data from the vine copula, following the procedure from Section 2.4. Finally, we use the decoder to transform the samples from vine in latent space into simulated images in pixel space. A schematic representation of this idea is given in Figure 1 and pseudo-code for the VCAE algorithm can be found in Appendix B.

The vine copula is fitted post-hoc for two reasons. First, since the nonparametric estimator is consistent for (almost) any distribution, the only purpose of the AE is to minimize the reconstruction error. The AE's latent space is unconstrained and the same AE can be used for both conditional and unconditional sampling. Second, it is unclear how to train a model that includes a nonparametric estimator since it has no parameters, there is no loss function to minimize or gradients to propagate. One possibility would be using spline estimators, which would allow to train the model end-to-end by fitting the basis expansion's coefficients. However, spline estimators of copula densities have been empirically shown to have inferior performance than the transformation kernel estimator [55].

There is some leeway in modeling choices related to the vine. For instance, the number of trees as well as the choice of copula family (i.e., Gaussian or nonparametric) have an impact of the synthetic samples, as sharper details are expected from more flexible models. Note that one can adjust the characteristics of the vine until an acceptable fit of the latent features even after the AE is trained.

## 4 Experiments

To evaluate VCAEs as generative models, we follow an experimental setup similar as related works on GANs and VAEs. We compare vanilla VAEs to VCAEs using the same architectures, but replacing the variational part of the VAEs by vines to obtain the VCAEs. From the generative adversarial framework, we compare to DCGAN [62]. The architectures for all networks are described in Appendix D.

Additionally, we explore two modifications of VCAE, (i) Conditional VCAE, that is sampling from a mixture obtained by fitting one vine per class label, and (ii) DEC-VCAE, namely adding a clustering-related penalty as in [81]. The rationale behind the clustering penalty was to better disentangle the features in the latent space. In other words, we obtain latent representations where the different clusters (i.e., classes) are better separated, thereby facilitating their modeling.

### 4.1 Experimental setup

#### Datasets and metrics

We explore three real-world datasets: two small scale - MNIST [40] and Street View House Numbers (SVNH) [57], and one large scale - CelebA [44]. While it is generally common to evaluate models by comparing their log-likelihood on a test dataset, this criterion is known to be unsuitable to evaluate the quality of sampled images [75]. As a result, we use an evaluation framework recently developed for GANs [82]. According to [82], the most robust metrics for two sample testing are the *classifier two sample test* (C2ST, [46]) and *mean maximum discrepancy* score (MMD, [23]). Furthermore, [82] proposes to use these metrics not only in the pixel space, but over feature mappings in convolution space. Hence, we also compare generative models in terms of Wasserstein distance, MMD score and C2ST accuracy over ResNet-34 features. Additionally, we also use the *common inception score* [66] and *Fréchet inception distance* (FID, [28]). For all metrics, lower values are better, except for inception. We refer the reader to [82] for further details on the metrics and the implementation.

#### Architectures, hyperparameters, and hardware

For all models, we fix the AE's architecture as described in Appendix D. Parameters of the optimizers and other hyperparameters are fixed as follows. Unless stated otherwise, all experiments were run with nonparametric vines and truncated after 5 trees. We use deep CNN models for the AEs in all baselines and follow closely DCGAN [62] with batch normalization layers for natural image datasets. For all AE-based methods, we use the Adam optimizer with learning rate 0.005 and weight decay 0.001 for all the natural image experiments, and 0.001 for both parameters on MNIST. For DCGAN, we use the recommended learning rate 0.0002 and $\beta_1 = 0.5$ for Adam. The size of the latent spaces $z$ was selected depending on the dataset's size and complexity. For MNIST, we present results with $z = 10$, SVHN $z = 20$ and for CelebA $z = 100$. We chose to present the values that gave reasonable results for all baselines. For MNIST, we used batch size of 128, for SVHN 32, and for CelebA batches of 100 samples for training. All models were trained on a separate train set, and evaluated on hold out test sets of 2000 samples, which is the evaluation size used in [82]. We used Pythorch 4.1 [58], and we provide our code in Appendix E. All experiments were executed on an AWS instance *p2.xlarge* with an NVIDIA K80 GPU, 4 CPUs and 61 GB of RAM.

### 4.2 Results

#### MNIST

In Figure 4, we present results from VCAE to understand how different copula families impact the quality of the samples. The independence copula corresponds to assuming independence between the latent features as in VAEs. And the images generated using nonparametric vines seem to improve over the other two. Within our framework, the training of the AE and the vine fit are independent. And we can leverage this to perform conditional sampling by fitting a different vine for each class of digit. We show results of vine samples per digit class in Figure 4.

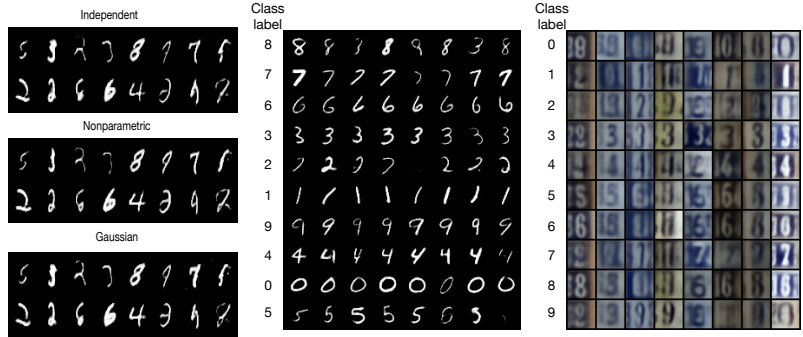

Figure 4: Left - impact of copula family selection on **MNIST**. Middle and Right - random samples of Conditional *VCAE* on **MNIST** and **SVHN**.

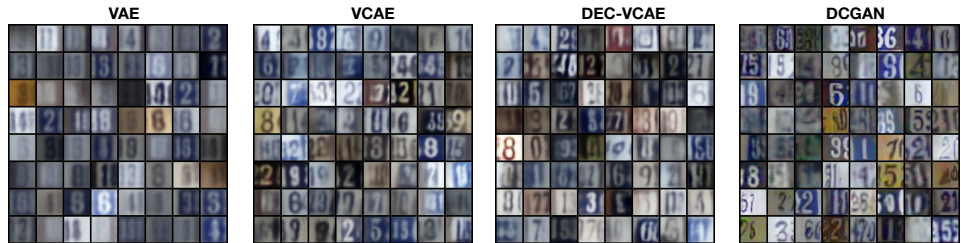

Figure 5: Left to right, random samples of *VAE*, *VCAE*, *DEC-VCAE*, and *DCGAN* for **SVHN**.

## SVHN

The results in Figure 5 show that the variants of vine generative models visually provide sharper images than vanilla VAEs when architectures and training hyper-parameters are the same for all models. All AE-based methods were trained on latent space $z = 20$ for 200 epochs, while for DCGAN we use $z = 100$ and evaluate it at its best performance (50 epochs). In Figure 6, we can see that VCAE and DEC-VCAE have very similar and competitive results to DCGAN (at its best) across all metrics, and both clearly outperform vanila VAE. Finally, the FID score calculated with regards to $10^4$ real test samples are has $0.205$ for VAE, $0.194$ for DCGAN and $0.167$ for VCAE which shows that VCAE also has slight advantage using this metric. In Appendix C.2, Figure 12 and Figure 13 show similar results respectively for the MNIST and CelebA datasets.

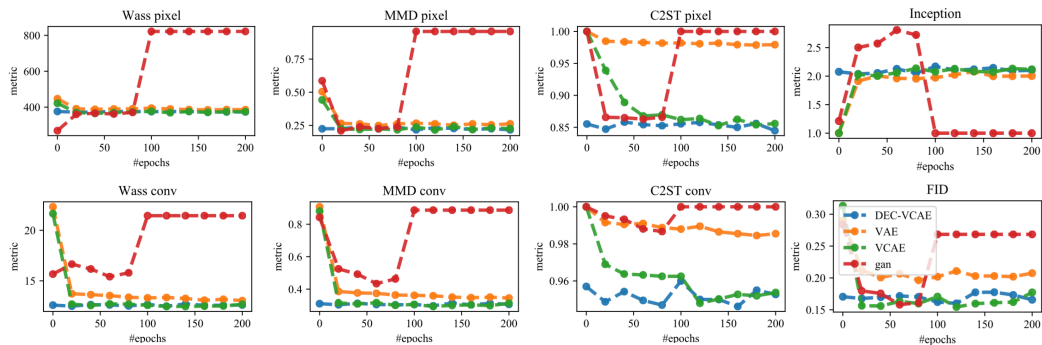

Figure 6: Various evaluation scores for all baselines on the **SVHN** dataset.

## CelebA

In the large scale setting, we present results for VCAE, VAE, and DCGAN only, because our GPU ran out of memory on DEC-VCAE. From the random samples in Figure 7, we see that, for the same amount of training (in terms of epochs), VCAE results is not only sharper but also produce more diverse samples. VAEs improve using additional training, but vine-based solutions achieve better results with less resources and without constraints on the latent space. Note that, in Appendix C.3, we also study the quality of the latent representation.

To see the effect of the number of trees in the vine structure, we include Figure 8, where we can see that from the random sample the vine with five trees provides images with sharper details.

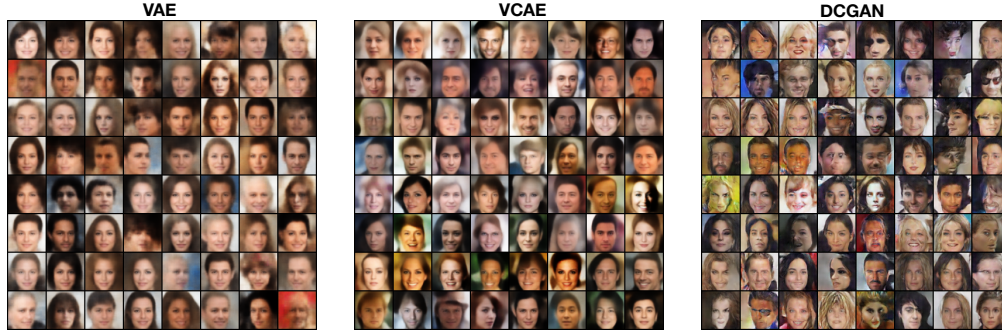

Figure 7: Random samples for models trained on the **CelebA** dataset, for *VAE* and *VCAE* at 200 epochs, and for *DCGAN* best results at 30 epochs.

Since, as stated in Section 2.3 and Section 2.4, the algorithms complexity increases linearly with the number of trees, we explore the trade-off between computation time and quality of the samples in Appendix C.4. Results show that, as expected, deeper vines, and hence longer computation times, improve the quality of the generated images. Finally, as for SVHN, the FID score shows an advantage of the vine-base method over VAEs as we find 0.247 for VAE and 0.233 for VCAE. For DCGAN the FID score is

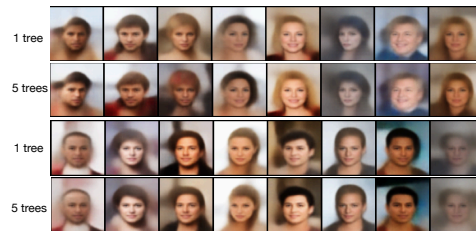

Figure 8: Higher truncation - sharper images.

0.169 which is better than VCAE, however, looking at the random batch samples in Figure 7 although GANs outputs sharper images, it is clear that VCAE produces more realistic faces.

**Execution times**

We conclude the experimental section with Table 1 comparing execution times. We note that VCAE compares favorably to VAE, which is a "fair" observation given that the architectures are alike. Comparison to DCGAN is more difficult, due to the different nature of the two frameworks (i.e., based respectively on AEs or adversarial).

Table 1: Execution times.

|  | MNIST (200 epochs) | SVHN (200 epochs) | CelebA (100 epochs) |
|---|---|---|---|
| **VAE** | 50 min | 4h 7 min | 7h |
| **VCAE** | 55 min | 1h 32 min | 6.5h |
| **DEC VCAE** | 101 min | 2h 35 min | / |
| **DCGAN** | 120 min (40 epochs) | 3h 20 min (50 epochs) | 5h (30 epochs) |

It should also be noted that the implementation of VCAE is far from optimal for two reasons. First, we use the R interface to `vinecopulib` in `Python` through `rpy2`. As such, there is a communication overhead resulting from switching between R and `Python`. Second, while `vinecopulib` uses native `C++11` multithreading, it does not run on GPU cores. From our results, this is not problematic, since the execution times are satisfactory. But VCAE could be much faster if nonparametric vines were implemented in a tensor-based framework.

## 5  Conclusion

In this paper, we present vine copula autoencoders (VCAEs), a first attempt at using copulas as high-dimensional generative models. VCAE leverage the capacities of AEs at providing compressed representations of the data, along with the flexibility of nonparametric vines to model arbitrary probability distributions. We highlight the versatility and power of vines as generative models in high-dimensional settings with experiments on various real datasets. VCAEs results show that they are comparable to existing solutions in terms of sample quality, while at the same time providing straightforward training along more control over flexibility at modeling and exploration (tuning truncation level, selection of copula families/parameter values). Several directions for future work and extensions are being considered. First, we started to experiments with VAEs having flexible distributional assumptions (i.e., by using a vine on the variational distribution). Second, we plan on studying hybrid models using adversarial mechanisms. In related work [38] (see Appendix F), we have also investigated the method's potential for sampling *sequential data* (artificial mobility trajectories). There can also be extensions to text data, or investigating which types of vines synthesize best samples for different data types.

## Footnotes

[1] A copula is a distribution function with uniform margins.

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
