[Supplementary Material · paper_with_appendix.pdf]

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

[2]$\mathbb{P}[U \leq u] = \mathbb{P}[F_X(X) \leq u] = \mathbb{P}[X \leq F_X^{-1}(u)] = F_X(F_X^{-1}(u)) = u$

[3]Coverage measures the probability mass of the true data covered by the approximate density of the learned model as $C := \mathbb{P}_{\text{data}}[d\mathbb{P}_{\text{model}} > t]$ where $t$ is selected such that $\mathbb{P}_{\text{model}}[d\mathbb{P}_{\text{model}} > t] = \alpha$ and where $d\mathbb{P}_{\text{model}}$ denotes the model density function. We set $\alpha = 0.95$ as in the original paper.

[4]reducing the learning rate by 10 after 30th epoch

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

# Appendix

## A   Introduction to (vine) copulas

### A.1   Copulas

Recall that the components of the random vector $(X_1, \ldots, X_d)$ are said to be independent if and only if its joint distribution $F$ is given by the product of the $d$ marginals $F_i$ for $i \in \{1, \ldots, d\}$, that is

$$F(x_1, \ldots, x_d) = \prod_{i=1}^{d} F_i(x_i), \tag{3}$$

for any $(x_1, \ldots, x_d) \in \mathcal{R}^d$. If the random variables are absolutely continuous, then differentiating (3) with respect to $(x_1, \ldots, x_d)$ implies that a similar statement hold for the densities, that is

$$f(x_1, \ldots, x_d) = \prod_{i=1}^{d} f_i(x_i), \tag{4}$$

where $f$ is the joint density, and $f_i$ for $i \in \{1, \ldots, d\}$ are the marginal densities.

However, when the variables are dependent, this statement is no longer true. In this case, the celebrated Sklar's theorem (see Theorem 1 for the precise statement) says that the joint distribution can be written as

$$F(x_1, \ldots, x_d) = C(F_1(x_1), \ldots, F_d(x_d)), \tag{5}$$

where $C$ is a *copula* that acts as a coupling mechanism between the $d$ marginals.

**Definition 1.**   *A $d$-dimensional copula copula is a multivariate cumulative distribution function $C : [0, 1]^d \to [0, 1]$ for which all the marginal distributions are uniform.*

In other words, for $d = 2$, $C$ is a distribution such that $C(1, u) = C(u, 1) = u$ for any $u \in [0, 1]$. Note that the simplest copulas is arguably the independence copula, namely plugging $C(u_1, \ldots, u_d) = \prod_{i=1}^{d} u_i$ into (5) leads to (3).

An intuitive way to understand the copula corresponding to a given joint distribution $F$ and marginal distributions $F_i$ for $i \in \{1, \ldots, d\}$ is as the distribution of the so-called probability integral transform (PIT) of the marginals.

**Definition 2.**   *The probability integral transform (PIT) of a random variable $X$ with distribution $F_X$ is the random variable $U = F_X(X)$.*

Because the PIT of any random variable is uniformly distributed[2], the joint distribution of the vector of PITs $(U_1, \ldots, U_d)$ with $U_i = F_i(X_i)$ for $i \in \{1, \ldots, d\}$ is a copula, namely $C$. A similar idea has an important consequence when one aims at sampling from the joint distribution $F$. Because it is well known that, if $U \sim \mathcal{U}[0, 1]$ and $F_X^{-1}$ is the inverse cumulative distribution of $X$, then $F_X^{-1}(U) \sim X$, transforming samples from $C$ into samples from $F$ is straightforward: if $(U_1, \cdots, U_d) \sim C$, then $X_i = F_i^{-1}(U_i)$ for $i \in \{1, \ldots, d\}$ implies that $(X_1, \ldots, X_d) \sim F$. While it looks like simply transforming a $d$-dimensional sampling problem into another $d$-dimensional sampling problem, vine copulas represent a model class for $C$ that is flexible and yet easy to sample from.

Viewing any joint distribution through this copula lens further yields a useful factorization: differentiating (5) with respect to $(x_1, \ldots, x_d)$ leads to

$$f(x_1, \ldots, x_d) = \frac{\partial^d F(x_1, \ldots, x_2)}{\partial x_1 \cdots \partial x_d} = \frac{\partial^d C(u_1, \ldots, u_d)}{\partial u_1 \cdots \partial u_d} \prod_{i=1}^{d} \frac{\partial F_i(x_i)}{\partial x_i} = c(u_1, \ldots, u_d) \prod_{i=1}^{d} f_i(x_i), \tag{6}$$

where $c$ is the so-called *copula density*, and $u_i = F_i(x_i)$ for $i \in \{1, \ldots, d\}$. Hence, we can see that the joint density factorize into a product between the marginal densities, similarly as in (4), with the copula density, which encodes the dependence. Taking the logarithm on both sides of (6), one obtains

$$\log f(x_1, \ldots, x_d) = \log c(u_1, \ldots, u_d) + \sum_{i=1}^{d} \log f_i(x_i).$$

Figure 9: Copula estimation. By row - (top) independent, (bottom) dependent variables. By column - (left) original data, (middle) pseudo-data after PIT, (right) estimated copula density.

In other words, the factorization implies that the joint log-likelihood is the sum of the marginal log-likelihoods and the copula log-likelihood. This observation can be conveniently leveraged for estimation via a two-step procedure where $f_i$ is first estimated by $\widehat{f_i}$ for $i \in \{1, \ldots, d\}$. Then, pseudo-observations of the copula are recovered using the estimated PITs, that is $u_i \approx \widehat{F_i}(x_i)$ for $i \in \{1, \ldots, d\}$, and $c$ is then estimated by $\widehat{c}$ using the pseudo-sample. This procedure is exemplified in Figure 9.

To summarize, copulas are a tool allowing to represent any multivariate distribution through the individual variables' marginal behaviors as well as their inter-dependencies. While lesser known in the machine learning community, copulas have been widely exploited by in other fields, from economics to quantitative finance, insurance and environmental sciences; in particular when capturing the joint tail behavior is of high importance. In financial risk management for instance, so-called tail events can trigger large and simultaneous losses (or gains) on portfolios. Consequently, multiple parametric copula families have been studied to capture lower/upper tail dependence, or no tail dependence at all. Similarly, other families have been developed to handle asymmetries or other dependence patterns. But such parametric families, which usually imply that the dependence between all pairs of variables is of the same kind, are seldom flexible enough in higher dimensions. Such limitations have led to the development of *pair-copulas constructions* (PCCs) or *vines* - hierarchical structures which allow to flexibly model high dimensional distributions by decomposing the dependence structure into *pairs of (bivariate) copulas*.

## A.2 Vines

According to [34, 4, 16], any copula density can be decomposed into a product of $\frac{d(d-1)}{2}$ bivariate (conditional) copula densities. While a decomposition is not unique, it can be organized as a graphical model, a sequence of $d - 1$ nested trees, called *regular vine*, *R-vine*, or simply *vine*. Denoting $T_m = (V_m, E_m)$ with $V_m$ and $E_m$ the set of nodes and edges of tree $m$ for $m = 1, \ldots, d-1$, the sequence is a vine if it satisfies the following set of conditions guaranteeing that the decomposition leads to a *valid joint density*:

- $T_1$ is a tree with nodes $V_1 = \{1, \ldots, d\}$ and edges $E_1$.
- For $m \geq 2$, $T_m$ is a tree with nodes $V_m = E_{m-1}$ and edges $E_m$.
- (Proximity condition) Whenever two nodes in $T_m + 1$ are joined by an edge, the corresponding edges in $T_m$ must share a common node.

The corresponding tree sequence is then called the *structure* of the PCC and has important implications to design efficient algorithms for the estimation and sampling of such models.

Each edge $e$ is associated to a bivariate copula $c_{j_e,k_e|D_e}$ (a so-called *pair-copula*), with the set $D_e \in \{1, \cdots, d\}$ and the indices $j_e, k_e \in \{1, \cdots, d\}$ forming respectively its *conditioning set* and the *conditioned set*. Finally, the joint copula density can be written as the product of all pair-copula densities

$$c(u_1, \cdots, u_d) = \prod_{m=1}^{d-1} \prod_{e \in E_m} c_{j_e,k_e|D_e}(u_{j_e|D_e}, u_{k_e|D_e}), \tag{7}$$

where

$$u_{j_e|D_e} = \mathbb{P}\left[U_{j_e} \leq u_{j_e} \mid \boldsymbol{U}_{D_e} = \boldsymbol{u}_{D_e}\right],$$

and similarly for $u_{j_e|D_e}$, with $\boldsymbol{U}_{D_e} = \boldsymbol{u}_{D_e}$ understood as component-wise equality for all components of $(U_1, \ldots, U_d)$ and $(u_1, \ldots, u_d)$ included in the conditioning set $D_e$. In Example Example 1 we present a full example of an R vine for a 5 dimensional density.

**Example 1.** *The density of a PCC corresponding to the tree sequence in Figure 10 is*

$$c = c_{1,2}\, c_{1,3}\, c_{3,4}\, c_{3,5}\, c_{2,3|1}\, c_{1,4|3}\, c_{1,5|3} c_{2,4|1,3}\, c_{4,5|1,3}\, c_{2,5|1,3,4}, \tag{8}$$

*where the colors correspond to the edges $E_1$, $E_2$, $E_3$, $E_4$.*

Figure 10: A vine tree sequence: the numbers represent the variables, $x, y$ the bivariate distribution of $x$ and $y$, and $x, y|z$ the bivariate distribution of $x$ and $y$ conditional on $z$. Each edge corresponds to a bivariate pair-copula in the PCC.

To summarize this section, in order to construct a vine, one has to choose two components:

- The structure, namely the set of trees $T_m = (V_m, E_m)$ for $m = 1, \ldots, d-1$.
- The pair-copulas, namely the models for $c_{j_e,k_e|D_e}$ for $e \in E_m$ and $m = 1, \ldots, d-1$.

To fix ideas, it is easier to start by assuming the structure to be known.

### A.2.1 Estimating the pair-copulas

To answer how one could estimate the pair-copulas is closely related whether one can evaluate the density in (7): if one can evaluate the density, then taking it's logarithm and finding the MLE would be straightforward. While it would be impractical for high-dimensional data, the factorization as a product of pair-copulas paves the way for a sequential procedure. Indeed, taking the logarithm of both sides of (7), we have

$$\log c(u_1, \cdots, u_d) = \sum_{m=1}^{d-1} \sum_{e \in E_m} \log c_{j_e,k_e|D_e}(u_{j_e|D_e}, u_{k_e|D_e}). \tag{9}$$

One can thus use (9) to proceed in a tree-wise fashion, starting with $m = 1$, with all pairs in a given tree, that is $e \in E_m$, being estimated in parallel.

Assuming the marginal distributions to be known, one can simply proceed with pseudo-observations $(U_1, \cdots, U_d)$ with $U_i = F_i(X_i)$ to estimate the pairs in the first tree (i.e., when $m = 1$). It works because, for those pairs, the conditioning set is empty, that is $D_e = \emptyset$. When the marginal distributions are unknown, one can proceed similarly using $\widehat{F}_i(X_i)$. But for the higher trees (i.e., when $m > 1$), the decomposition involves conditional distributions like $U_{j_e}|\boldsymbol{U}_{D_e}$ with a non-empty conditioning set, that is $D_e \neq \emptyset$.

It turns out that the arguments for pair-copulas in any tree $m > 1$ can be expressed recursively using conditional distributions corresponding to bivariate copulas in the previous tree (i.e., $m - 1$) as follows. Let $e \in E_m$ be an edge of tree $m$ and $l_e \in D_e$ be another index such that $c_{j_e, l_e | D_e \setminus l_e}$ is a pair-copula in tree $m - 1$, and define $D_e' = D_e \setminus l_e$. Then we have that

$$u_{j_e | D_e} = h_{j_e, l_e | D_e'}(u_{j_e | D_e'}, u_{l_e | D_e'})$$

where the so-called $h$-function is defined as

$$h_{j_e, l_e | D_e'}(u_1, u_2) := \int_0^{u_1} c_{j_e, l_e | D_e'}(v, u_2) dv = \frac{\partial C_{j_e, l_e | D_e'}(u_1, u_2)}{\partial u_2}.$$

In each step of this recursion the conditioning set $D_e$ is reduced by one element, until we eventually reach the first tree with $D_e = \emptyset$. Note that, in a vine, for any edge $e$, the existence of an index $l_e$ such that $c_{j_e, l_e | D_e \setminus l_e}$ is a pair-copula in tree $m - 1$ is guaranteed. This allows us to write any of the required conditional distributions as a recursion over $h$-functions that directly linked to the pair-copula densities in previous trees. As such, assuming the structure to be known, a sequential algorithm to estimate the pair-copulas can be described as follow:

1. Set $m = 1$ and estimate all pair-copulas for the first tree using $(U_1, \cdots, U_d)$.
2. Set $m = m + 1$ and compute the conditional distributions $u_{j_e | D_e}$ and $u_{k_e | D_e}$ for $e \in E_m$.
3. Estimate all pair-copulas in tree $m$ using $u_{j_e | D_e}$ and $u_{k_e | D_e}$ for $e \in E_m$.
4. If $m = d - 1$, all pairs have been estimated. Otherwise, go to step 2.

The procedure is generic in the sense that it can be used with any bivariate copula estimator, and we refer to Algorithm 1 in [54] for its pseudocode. Note that the decomposition can also be truncated by replacing the termination condition at step 4 using any truncation level smaller than $d - 1$. Finally, for each pair-copula, one could also estimate different models at step 3 and select the best one according to some suitable criterion (e.g., AIC or BIC). One important question that we brushed aside is: given that the structure is generally unknown, how can we also select it?

### A.2.2 Selecting the structure

To learn the structure for a dataset where it is unknown, multiple solutions have been proposed. In this paper, as it is most common in the vine literature, we use the so-called Dissmann algorithm, first proposed in [19]: This algorithm represents a greedy heuristic aiming at capturing higher dependencies in the lower trees. The intuition is that higher-tree represent higher-order interactions, which are harder to estimate. As such, one should prioritize modeling the most important patterns in lower trees. This is achieved by finding the maximum spanning tree (MST) using a dependence measure as edge weights. For instance, the absolute value of the empirical Kendall's $\tau$ for monotone dependencies or the maximal correlation [63] for more general patterns are popular choices. To compute the MST, most implementations use Prim's algorithm [60]. Letting $\tau$ denote a generic bivariate dependence measure, the sequential algorithm mentioned above can thus be modified in a straightforward manner:

1. Set $m = 1$ and compute the dependence $\tau(U_i, U_j)$ for all pairs $1 \leq i < j \leq d$. While this defines a complete graph, only keep the edges corresponding to the MST in $E_m$. Finally, estimate all pair-copulas for the first tree as before.
2. Set $m = m + 1$ and compute the conditional distributions $u_{j_e | D_e}$ and $u_{k_e | D_e}$, as well as the dependence $\tau(u_{j_e | D_e}, u_{k_e | D_e})$ for all pairs where $e$ is an edge allowed by the proximity condition. Only keep the edges corresponding to the MST in $E_m$.
3. Estimate all pair-copulas in tree $m$ using $u_{j_e | D_e}$ and $u_{k_e | D_e}$ for $e \in E_m$.
4. If $m = d - 1$, all pairs have been estimated. Otherwise, go to step 2.

Note that step 2 can be implemented efficiently by observing that, while conditional distributions might appear in multiple candidate edges, they can be computed only once and stored for further use. The resulting estimation and structure selection procedure is summarized in Algorithm 2 of [54].

### A.3 Assumptions for the consistency and asymptotic normality of the kernel bivariate copula estimator

(B1) $\partial_u C(u, v)$ and $\partial_{uu} C(u, v)$ exist and are continuous on $(u, v) \in (0, 1) \times [0, 1]$, and there exists a constant $Q_1$ such that $|\partial_{uu} C(u, v)| \leq Q_1 / u(1 - u)$ for $(u, v) \in (0, 1) \times [0, 1]$.

(B2) $\partial_v C(u,v)$ and $\partial_{vv} C(u,v)$ exist and are continuous on $(u,v) \in [0,1] \times (0,1)$, and there exists a constant $Q_2$ such that $|\partial_{vv} C(u,v)| \leq Q_2/v(1-v)$ for $(u,v) \in [0,1] \times (0,1)$.

(B3) The density $c(u,v) = \partial_{uv} C(u,v)$ admits continuous second-order partial derivatives in $(0,1)^2$ and there exists a constant $Q_0$ such that, for $(u,v) \in (0,1)^2$, $c(u,v) \leq Q_0 \min\left(\frac{1}{u(1-u)}, \frac{1}{v(1-v)}\right)$.

# B  The VCAE algorithm

The algorithm for vine copula autoencoders is given in Algorithm 1.

---
**Algorithm 1** Vine Copula Autoencoder
---

**Input:** train set $X$ of $\{x_1, x_2, ...x_n\}$ images.
1. Train AE component with $X$:
$f \leftarrow encoder$
$g \leftarrow decoder$

**2**. Encode train set with $f$ :
$\phi(X) \leftarrow f(X)$

**3**. Fit a vine copula $c$ using encoded features:
$c \leftarrow \{\phi_1, \phi_2, ...\phi_n\}$ (as described in Section 2.2 and Section 2.3).

**4**. Sample random observations form $c$:
$\phi' \leftarrow c(\phi)$ (as in Section 2.4)

**5**. Decode the random features:
$X' \leftarrow g(\phi')$

**Output:** generated images $X'$.

---

## B.1  Variations of VCAE

**Conditional VCAE**  Since the vine estimation and the AE training are independent in our approach, we can do steps 3–5 in Algorithm 1 per class label (fit a vine per class feature) which makes the implementation of Conditional VCAE straightforward.

**DEC-VCAE**  For the implementation of the DEC-VCAE we followed the instructions from the authors in [81]. A difficulty with AEs is that the encoded features are typically entangled, even when the AE reconstruction is accurate. Therefore we enforce some clustering. We start with an pre-trained AE and then optimize a two-term loss function: the clustering and the reconstruction loss.

# C  Additional experiments

## C.1  Toy datasets

Similarly to related generative model literature [26, 77], we test our method on two-dimensional toy datasets. Since this is a 2D case, we use bivariate copulas with nonparametric marginal densities for the estimation and sampling. The three datasets are ring of isotropic Gaussians with 8 modes, $5 \times 5$ grid of isotropic Gaussians and the swiss roll dataset. These datasets have proven to be challenging for GANs due to the mode collapse issues [26, 77]. They motivate how the flexibility of nonparametric copulas can be leveraged, and we additionally compare to a baseline Gaussian copula. From Figure 11, we observe the benefits of using nonparametrics; while fitting such datasets is easy, it is clear that the Gaussian assumption is not suitable in such cases (except for the grid of Gaussians).

We further confirm this quantitatively in Table 2, where we repeat the experiment on 100 random datasets of each type, and present the average and standard deviations for both copula families. To evaluate the sampled images, additionally to the MMD, we use the negative log-likelihood (NLL)

Table 2: Evaluation on toy datasets for nonparametric and Gaussian copula. Average and standard deviations from 100 repetitions.

|  | Ring | Grid | Swiss roll |
|---|---|---|---|
| **nonparametric** | | | |
| **NLL** ↑ | -2.47(0.15) | -3.77(0.2) | -5.23(0.05) |
| **Coverage** ↑ | 0.93(0.02) | 0.94(0.02) | 0.99(0.01) |
| **MMD** ↓ | 0.18(0.02) | 0.15(0.16) | 0.32(0.03) |
| **Gaussian** | | | |
| **NLL** ↑ | -2.98(0.05) | -3.34(0.07) | -6.21(0.05) |
| **Coverage** ↑ | 0.95(0.02) | 0.96(0.014) | 0.93(0.03) |
| **MMD** ↓ | 0.33(0.02) | 0.14(0.02) | 0.38(0.02) |

Figure 11: Copula generated data - top row nonparametric, bottom row Gaussian copula.

and *coverage*[3], a closely related metric [77]. As expected, nonparametrics provide better samples according to the three two-sample metrics.

## C.2   Various evaluation metrics for the MNIST and CelebA dataset

In Figure 12 and Figure 13, we present the evaluation scores for all baselines on the **MNIST** and **CelebA** datasets. Note that, in the evaluation framework that we use [82], the Inception Score and FID are based on ImageNet features. Therefore those scores are not suitable for binary images and excluded from Figure 12.

The results in Figure 13 show that DCGAN has a slight advantage over VAE and VCAE when methods are evaluated in feature space, while VCAE outperforms VAE on all metrics. In this experiment we used adaptive learning rate for DCGAN [4] to evaluate the scores on more than 30 epochs.

Figure 12: Various evaluation scores for all baselines on the **MNIST** dataset.

Figure 13: Various evaluation scores for all baselines on the **CelebA** dataset.

## C.3 Interpolation in latent space

Figure 14 shows that the transitions for VCAE are smooth and without any sharp changes or unexpected samples in-between when walking the latent space by linear interpolation between two test samples as in [62]. This is not explicitly related to VCAE generative models since we do not train an end-to-end model, however it is important to show that the AE network we use did not simply memorize images.

Figure 14: Interpolation in latent space between two real samples (shown in the first two columns) with a VCAE trained on **CelebA**

## C.4 The trade-off between time complexity and sample quality

To explore the effect of the choice for truncation level, i.e. the depth of the vine (number of trees) over the quality of the produced VCAE samples, we include an ablation study on the FashionMNIST dataset [80]. The quantitative and qualitative evaluation in Figure 15 and Figure 16 suggest that higher level of truncation provide better samples, at the expected cost of longer computation times.

Figure 15: Quantitative evaluation of various truncation levels for the VCAE on **FashionMNIST**.

Figure 16: Qualitative evaluation of various truncation levels (left panel) and computation time with respect to the vine depth (right panel) for the VCAE on **FashionMNIST**.

# D  Additional details on the experiments

We use the same AE architecture for VCAE, DEC-VCAE and VAE as described below. All the AEs were trained by minimizing the Binary Cross Entropy Loss.

## D.1  MNIST

The only transformation performed on this dataset is a padding of 2. By doing so we are able to use the same architecture for multiple datasets. We use CNNs for the encoder and the decoder whose architectures are as follows:

- Encoder:

$$
\begin{aligned}
x \in R^{32 \times 32} &\to Conv_{32} \to BN \to ReLU \\
&\to Conv_{64} \to BN \to ReLU \\
&\to Conv_{128} \to BN \to ReLU \\
&\to FC_{10}
\end{aligned}
$$

- Decoder:

$$
\begin{aligned}
z \in R^{10} \to FC_{100} &\to ConvT_{128} \to BN \to ReLU \\
&\to ConvT_{64} \to BN \to ReLU \\
&\to ConvT_{128} \to BN \to ReLU \\
&\to FC_1
\end{aligned}
$$

- DCGAN Generator:

$$
\begin{aligned}
z \in R^{100} \to\to &ConvT_1 \to BN \to ReLU \\
&\to ConvT_{128} \to BN \to ReLU \\
&\to ConvT_{64} \to BN \to ReLU \\
&\to ConvT_{32} \to BN \to ReLU \\
&\to ConvT_{16} \to BN \to ReLU \\
&\to Tanh_1
\end{aligned}
$$

- DCGAN Discriminator:

$$
\begin{aligned}
Conv_1 &\to BN \to LeakyReLU \\
\to Conv_{16} &\to BN \to LeakyReLU \\
\to Conv_{32} &\to BN \to LeakyReLU \\
\to Conv_{64} &\to BN \to LeakyReLU \\
\to Conv_{128} &\to BN \to LeakyReLU \\
&\to Sigmoid_1
\end{aligned}
$$

with all (de)convolutional layers have $4 \times 4$ filters, a stride of 2, and a padding of 1. We use BN to denote batch normalization and ReLU for rectified linear units and FC for fully connected layers. We denote $Conv_k$ the convolution with $k$ filters. Leaky ReLU was used with negative slope = 0.2 everywhere.

## D.2  SVHN

For SVHN we use the data as is without any preprocessing. The architectures are:

- Encoder:

$$
\begin{aligned}
x \in R^{3 \times 32 \times 32} &\to Conv_{64} \to BN \to LeakyReLU \\
&\to Conv_{128} \to BN \to LeakyReLU \\
&\to Conv_{256} \to BN \to LeakyReLU \\
&\to FC_{100} \to FC_{20}
\end{aligned}
$$

- Decoder:

$$z \in R^{20} \to FC_{100} \to ConvT_{256} \to BN \to ReLU$$
$$\to ConvT_{128} \to BN \to ReLU$$
$$\to ConvT_{64} \to BN \to ReLU$$
$$\to ConvT_{32} \to BN \to ReLU$$
$$\to FC_1$$

- DCGAN Generator:

$$z \in R^{100} \to ConvT_{256} \to BN \to ReLU$$
$$\to ConvT_{128} \to BN \to ReLU$$
$$\to ConvT_{64} \to BN \to ReLU$$
$$\to ConvT_{32} \to BN \to ReLU$$
$$\to ConvT_3 \to BN \to ReLU$$
$$\to Tanh_1$$

- DCGAN Discriminator:

$$Conv_3 \to BN \to LeakyReLU$$
$$\to Conv_{32} \to BN \to LeakyReLU$$
$$\to Conv_{64} \to BN \to LeakyReLU$$
$$\to Conv_{128} \to BN \to LeakyReLU$$
$$\to Conv_{256} \to BN \to LeakyReLU$$
$$\to Sigmoid_1$$

where all (de)convolutional the layers have $4 \times 4$ filters, a stride of 2, and a padding of 1. The rest of the notations are the same as before.

## D.3 CelebA

For CelebA we first took central crops of $140 \times 140$ and then resized to resolution $64 \times 64$. Note that only Fig. 9 in the main text is not a result of this preprocessing. The architectures used are as follows:

- Encoder:

$$x \in R^{3 \times 64 \times 64} \to Conv_{64} \to BN \to LeakyReLU$$
$$\to Conv_{128} \to BN \to LeakyReLU$$
$$\to Conv_{256} \to BN \to LeakyReLU$$
$$\to Conv_{512} \to BN \to LeakyReLU$$
$$\to FC_{100} \to FC_{100}$$

- Decoder:

$$z \in R^{100} \to FC_{100} \to ConvT_{512} \to BN \to ReLU$$
$$\to ConvT_{256} \to BN \to ReLU$$
$$\to ConvT_{128} \to BN \to ReLU$$
$$\to ConvT_{64} \to BN \to ReLU$$
$$\to ConvT_{32} \to BN \to ReLU$$
$$\to FC_1$$

- DCGAN Generator:

$$z \in R^{100} \to ConvT_{512} \to BN \to ReLU$$
$$\to ConvT_{256} \to BN \to ReLU$$
$$\to ConvT_{128} \to BN \to ReLU$$
$$\to ConvT_{64} \to BN \to ReLU$$
$$\to ConvT_3 \to BN \to ReLU$$
$$\to Tanh_1$$

- DCGAN Discriminator:

$$Conv_3 \rightarrow BN \rightarrow LeakyReLU$$
$$\rightarrow Conv_{64} \rightarrow BN \rightarrow LeakyReLU$$
$$\rightarrow Conv_{128} \rightarrow BN \rightarrow LeakyReLU$$
$$\rightarrow Conv_{256} \rightarrow BN \rightarrow LeakyReLU$$
$$\rightarrow Conv_{512} \rightarrow BN \rightarrow LeakyReLU$$
$$\rightarrow Sigmoid_1$$

where all the (de)convolutional layers have $4 \times 4$ filters, a stride of 2, and a padding of 1. Padding was set to 0 only for the last convoluitional layer of the encoder and the first layer of the decoder. The rest of the notations are the same as before.

## E    Code

Our code is available at the following link: https://github.com/tagas/vcae.

## F    Simulating Mobility Trajectories with copulas

In related work [38], we have also compared to adversarial and recurrent based methods for sampling *sequential data* (artificial mobility trajectories). We evaluate the generated trajectories with respect to their geographic and semantic similarity, circadian rhythms, long-range dependencies, training and generation time. We also include two sample tests to assess statistical similarity between the observed and simulated distributions, and we analyze the privacy trade-offs with respect to membership inference and location-sequence attacks. The results show that copulas surpass all baselines in terms of MMD score and training + simulation time. For more details please see [38].