[Reviews · NeurIPS 2019]

Reviewer 1



- Introduction: the way to the argument why copulas are a useful object for building AE architecture appears somewhat convoluted and indirect; please try to streamline the way towards the 'why' and 'why now' behind the project - this AE specification exploits that copulas allow to decompose a joint density into a product between the marginal densities and the dependence structure represented by the copula density - one aspect that may lead to misunderstanding: the approach is advertised as free of assumptions at the beginning, but then Gaussian copulas are implicated later...please clarify

Reviewer 2



The paper proposes to introduce pair-copula construction in the autoencoder architecture to create more robust generative model. Specifically, with a conventionally trained autoencoder encoding input data into a low dimensional latent space, the authors propose estimating the encoding vector distribution using vine-copulas. It is claimed that such estimation can be done efficiently based on sequential estimation of the pair copulation decomposition on vine trees. Furthermore, the estimated distribution can be sampled easily and passed to the decoder to create new data, thus serve as a generative model. My biggest issue with the work is the presentation, which needs a lot of improvements. In particular, the concept, models and algorithms, in general, are poorly presented or developed. The paper assumes an extensive background knowledge from its readers. While it is fair to assume certain technical background understanding on the topics, the paper should sufficiently summarize the core algorithms or concepts used from other referred works. This is far from the case here. The authors spend paragraphs explaining basic concepts such as fundamental architectures of autoencoders (Section 3), as well as very basic definitions of copula and sklar’s theorem. But rather than going into more details about many essential algorithms in the context of the paper, the authors chose to gloss over many of them or simply refer the reader to other works. In particular, more detailed description should cover, Model selection of copula Sequential estimation, constructing pseudo-observations (not defined throughout the paper) More detailed explanation of the nonparametric estimation Consequently, there is limited insight on various aspects of the proposed algorithm. How does the selection of copula affect the complexity and quality of the estimations and generated results Similarly for the truncation of the vine tree sequence (and independence assumptions) Continuity and differentiability conditions for the copula decomposition and estimation, how do they affect the model, if any? To the end, it looks like a very interesting approach to use copula decomposition in the center of autoencoder for generative models. The authors also claimed some advantage of VCAE against VAE and DCGAN, in terms of complexity, eyeballing test and a couple of metrics such as MMD and C2ST. But the paper was poorly presented and is lack of detailed analysis or insight. There are many typos and mistakes in the paper. Here is an incomplete list: Page 3 referred to Appendix A. 2 for “visual example of both the estimation and sampling of copula-based distributions. I don’t see the example at A.2 Section 2.2, for the edge e, D_e, j_e, k_e are generally subsets of the index set {1,...,d}, the authors used “element of” symbols. Typo, Section 2.5 “estimato” Section 2.4, “Furthermore, there exists” Figure 2: where is equation (7), you mean (2)? Figure 3 description, shouldn’t it be setting c_{2.3|1} to independence? Section 2.4, “Let C be a copula C”? After author rebuttals: I have read all the reviews and author rebuttals. I agree that the paper scores high on novelty/contribution, while the presentation lowers the quality considerably. I trust the authors will address the feedback, so I have increased my overall score to accept.

Reviewer 3



The proposed approach of combining copula and autoencoder as a generative model is quite novel. It has a superior performance to Variational Autoencoders and comparable to DCGAN ( one win and one loss) with modest gains in execution speed. Having said that, the comparison is based on only three datasets. Thus the results are not very conclusive. Perhaps there needs to be a better notation for the numerator of formula 2. The mean and covariance given as subscript appear weird. It took me some time to figure out thats what it means. In Figure 3, why does the graph for X1 versus X2 for C1 and C2 have circular contours? Shouldn't the contours be square since the model is nonparametric; X1,X2 have a uniform distribution over a square. I understand that copula help generate more diverse images, but I fail to understand why they give sharper images? Is there a good intuition for that? Figure 4: The caption says that the rightmost plot is a random sample of VCAE on MNIST and SVHN. How can there be a single sample for both SVHN and MNIST. I couldn't find graphs similar to Figure 6 for MNIST and CELEBA. Why were they omitted? The images in Figure 7 are too small to recognized the subtle points made by the author. It would be useful to have larger pictures in the supplement. How can one objectively verify that VCAE is not memorizing the pictures in Fig 7? I've seen section D on interpolation in the supplement. Is that enough to verify diversity? Please proofread the paper again; there are a few Typos and grammatical errors.

[Author Response · NeurIPS 2019]

**Reviewer #1**

- *Introduction.* We completely agree and will further emphasize that copulas repurpose a tool learning data representations into a full-fledged generative model. Copulas allow to easily (and at a comparatively small computational cost) turn any AE into a generative model with performances that compare favorably to state-of-the-art methods. We will rewrite the intro and highlight this key point, it currently appears only in lines 68 - 75 and in Appendix F.
- *Gaussian copula.* We apologize for the confusion. In (2), $\mathcal{N}_{(\Phi^{-1}(u_j),\Phi^{-1}(v_j)),\Sigma}$ is a bivariate Gaussian distribution with mean $(\Phi^{-1}(u_j),\Phi^{-1}(v_j))$ where $\{u_j,v_j\}_{j=1}^n$ are the observations. (2) defines a kernel estimator of the copula density and not a Gaussian copula. We only present Gaussian copulas in Figures 3 and 4 to show how the Gaussianity assumption results in worse synthetic samples compared to the nonparametric copulas. We will clarify this in the text.
- *Typos, restructuration, and clarifying captions.* Thank you for the comments, we agree and will correct as suggested.
- *Conclusion.* An executive summary of the empirical results is indeed lacking in the conclusion and will be added.

**Reviewer #2**

- *Contributions.* We agree that the paper is fairly practical rather than theoretical, but simple and simplistic should not be confused. Given the general interest in generative modeling, we feel like a method allowing to repurpose AEs into generative models at a small computational cost is worthy in itself.
- *Presentation of concepts/models/algorithms.* We will extend each topic in the supplementary to make the paper as self-contained as possible. But note that the pseudo-observations are already described lines 96-97 and in Figure 2, no model selection of copulas is required since only nonparametric pairs are used, sequential estimation is described over 11 lines while referencing to the rich literature on the topic, and nonparametric estimation takes about half a page. Basic concepts such as Sklar's theorem/copula definition take only 3 lines + 4 for the density (will cut 1/2), and AEs take 11 lines before switching to generative modeling (hard to cut). Vine copulas have generated thousands of papers in the last decade and, given the space constraint, we restricted ourselves to two pages (1/4 of the paper).
- *Copula selection.* As mentioned, no selection is required since (2) (i.e., nonparametric copulas) is used for every pair.
- *Complexity.* We will add to the paper that complexity $\approx O(n \times dim \times trunc\_lvl)$ for estimation/sampling algorithms, both involving a double loop over dimension/trunc level with an internal step scaling linearly with the sample size.
- *Quality of the samples and truncation.* We will add an extended analysis. See the figure below for preliminary results suggesting that deeper vines (i.e., longer computation times) improves the quality of the generated samples. Note also the linear scaling of computation time with truncation level.
- *Continuity/differentiability.* We will add the needed assumptions for the asymptotic properties of (2) as in [19].
- *Advantages over VAEs and GANs.* Due to space constraints, the analysis detailed the comments of lines 68-75 was moved to Appendix F. We will add it back (see a similar comment from Rev#1). Regarding complexity, our claim is simply that VCAEs are easier to train. We will also describe better the results from the different metrics.
- *Typos and mistakes.* Thank you for spotting them, we agree, will correct all, and proofread better. Page 3, Appendix A.2 should be Figure 2. $D_e$ is indeed a subset, but the other two are "elements of". Figures 2/3, yes and yes.

(A) Quantitative evaluation of various truncation levels on fashion MNIST  (B) Qualitative comparison  (C) Computation time wrt vine depth (fashion MNIST)

**Reviewer #3**

- *Conclusiveness of results.* This paper is a first-attempt at an alternative approach to seamlessly construct generative models by combining vines and AEs, and we chose arguably the three most common datasets and two best known competitors to illustrate our method's potential. The aim was neither an extensive empirical analysis, nor it was to prove definitive superiority against all state-of-the-art methods. In any case, Fashion MNIST will be added following the preliminary analysis from the figure above.
- *Formula 2.* Agreed, it also have been confusing to Rev#1, we could write something like $\mathcal{N}_{\mu,\Sigma}$ where $\mu = \dots$
- *Fig 3 contours.* We should have written the contours are presented in the kernel space, i.e. after transformation to Gaussian margins (lines 172-174) and circular because $X_1$ and $X_2$ are independent.
- *Intuition for sharper images.* Two potential explanations will be added. First, blurriness in VAEs comes from the independence and Gaussianity assumptions for the latent features, but we do not assume this. Second, adding depth (trees) to the vine structure results in more dependencies/details captured, and hence hence sharper images.
- *Fig 4.* Sorry for the mix-up, the middle and right panels correspond respectively to samples for MNIST and SVNH.
- *Fig 6 for MNIST and CelebA.* The experiments were not finished by the deadline but will be added to the final version. Preliminary results indicate that the conclusions will be similar as for SVNH.
- *Fig 7 and proofreading.* Thanks for noticing about Fig 7, and proofreading was also asked by Rev#2, our apologies.
- *Memorizing in Fig 7.* Since a vine is estimated on the latent features, memorization is rather the AE's issue. Because of the struggle in the community over how to tackle memorization evaluation (see e.g.Theis et al. , 2016, Borji 2018), we thought it to be out of the score of this work, but we will mention it.

[Meta-Review · NeurIPS 2019]

This paper proposes a vine copula autoencoder to construct flexible generative models for high-dimensional, structured data in three steps. By exploiting vine copulas, the proposed approach can transform any already trained autoencoder into a flexible generative model at a low computational cost, and its good performance was nicely demonstrated. This is a nice contribution to the field of constructing deep generative models.